



# Hygroscopic Properties of Aminium Sulphate Aerosols

Grazia Rovelli[1,2], Rachael E.H. Miles[1], Jonathan P. Reid[1], Simon L. Clegg[3]

[1] School of Chemistry, University of Bristol, Bristol, BS8 1TS, UK
[2] Department of Earth and Environmental Sciences, University of Milano-Bicocca, 20124 Milan, Italy
5  [3] School of Environmental Sciences, University of East Anglia, Norwich NR4 7TJ, UK

*Correspondence to*: Jonathan P. Reid (j.p.reid@bristol.ac.uk)

**Abstract.** Alkylaminium sulphates originate from the neutralisation reaction between short-chained amines and sulphuric acid and have been detected in atmospheric aerosol particles. Their physicochemical behaviour is less well characterised than their

inorganic equivalent, ammonium sulphate, even though they play a role in atmospheric processes such as the nucleation and growth of new particles and cloud droplet formation. In this work, a comparative evaporation kinetics experimental technique using a cylindrical electrodynamic balance is applied to determine the hygroscopic properties of six short-chained alkylaminium sulphates, specifically mono-, di- and tri-methylaminium sulphate and mono-, di- and tri-ethyl aminium sulphate.  This approach allows the retrieval of a water activity dependent growth curve in less than 10 s, avoiding the

uncertainties that can arise from the volatilisation of semi-volatile components. Measurements are made on particles >5 μm in radius, avoiding the need to correct equilibrium measurements for droplet surface curvature with assumed values of the droplet surface tension. Variations in equilibrium solution droplet composition with varying water activity are reported over the range 0.5 to >0.98, along with accurate parameterisations of solution density and refractive index. The uncertainties in water activities associated with the hygroscopicity measurements are typically <±0.2% at water activities >0.9 and ~±1% below 0.9, with

maximum uncertainties in diameter growth factors of ±0.7%. Comparison with previously reported measurements show deviation across the whole water activity range.



# 1. Introduction

Quantifying the response of aerosol particles to variations in relative humidity (RH) in the atmosphere is important for predicting the impact of aerosols on climate through both direct and indirect radiative forcings (Albrecht, 1989; Kanakidou et al., 2005; Kolb et al., 2010), for understanding the mechanisms of heterogeneous chemistry (Dennis-Smither et al., 2014; Lee et al., 2012) and the partitioning of products in the condensed phase (Dusek et al., 2006; Topping et al., 2013a), and for inferring the deposition and loss mechanism of aerosols on surfaces and on inhalation into the respiratory tract (Broday and Georgopoulos, 2001; Haddrell et al., 2015). With an increase in RH, solution droplets must absorb water to maintain an equilibrium balance of water between the gas and condensed phases, leading to the dilution of solutes, increase in mass of condensed phase water and hygroscopic growth in particle size. The hygroscopic response of a wide range of inorganic and organic solutes of varying solubility often present as complex mixtures, must be quantified. Thus, the influence of interactions between multiple solutes of varying charge, acidity and hydrophilicity must be understood through predictions of activity coefficients that reflect the departure of solution composition from ideality (Dutcher et al., 2013; Shiraiwa et al., 2013).

Measurements of the hygroscopic growth of ambient aerosol are routinely made: the extent of growth at high RH or the critical supersaturation for activation as cloud condensation nuclei are often represented by a single parameter, such as the κ parameter in κ-Köhler theory (Petters and Kreidenweis, 2007). Resorting to such a reduced parameter model is a consequence of the complex chemical composition of ambient aerosol and the intractability of providing accurate predictions based on a detailed chemical speciation. However, laboratory measurements are essential for exploring the complex details often obscured in measurements of ambient aerosol. These include the roles of pH, liquid-liquid phase separation and mixing state (Topping et al., 2013c; You et al., 2014) and the co-condensation of semi-volatile organic species with varying RH (Topping et al., 2013b). When considered alongside measurements of hygroscopic response and change in particle composition with RH for aerosol of well-known chemical composition, detailed predictive tools can be benchmarked and improved, providing a crucial framework for interpreting field measurements. We present here measurements of the hygroscopic response of a class of compounds, aminium sulphates, with the objective of providing accurate data to refine predictive tools of their equilibrium hygroscopic response.

Low molecular weight amines are mostly emitted into the atmosphere as gaseous compounds by a number of different sources, both anthropogenic (animal husbandry, food processing and cooking, combustion, pesticides) and natural (oceans, biomass burning, vegetation) (Ge et al., 2011a). These chemical species are basic (with pKa values from 9.8 to 10.84) (Lide, 2005), highly water-soluble and have high pure liquid vapour pressures (up to hundreds of kPa at 298.15 K) (Ge et al., 2011b). Their ambient concentrations in the environment can span wide ranges, depending on the sampling location (for example, up to 140 mg m$^{-3}$ close to a city market (Namieśnik et al., 2003), 110-300 ng m$^{-3}$ in the exhaust gas of a waste disposal (Kallinger and Niessner, 1999), of the order of tens of μg m$^{-3}$ inside livestock buildings (Kallinger and Niessner, 1999) and on the season




(Akyüz, 2008; Pratt et al., 2009). In the condensed phase, amines can account for hundreds of pg m⁻³ or a few ng m⁻³ of aerosol mass; an extensive review of the measured concentrations of a large number of amines both in the gas phase and in aerosols is given by Ge et al. (Ge et al., 2011a, 2011b).

Despite the volatility of short-chained alkylamines, these compounds can undergo gas-to-particle partitioning due to a variety of chemical processes (Ge et al., 2011a): direct solubilisation, oxidation reactions that lead to the formation of secondary organic aerosols, acid-base reactions similar to those of ammonia (Seinfeld and Pandis, 2006) with both inorganic (sulphuric, nitric and hydrochloric acids) and organic acids (Lavi et al., 2015; Liu et al., 2012), and displacement reactions of ammonium cations with aminium cations (Bzdek et al., 2010; Qiu and Zhang, 2013). With respect to these last two types of chemical

reactions, aminium sulphates are the products of the neutralisation of sulphuric acid and short-chained alkylamines:

$$H_2SO_4 + 2R_3N \rightleftarrows (R_3NH)_2SO_4 \qquad\qquad\qquad (1)$$

where R can be either –CH₃, –CH₂CH₃ or –H. The formation of aminium sulphates and other similar aminium salts increases their solubility and, thus, their partitioning from the gas to the condensed phase (Barsanti et al., 2009; Yli-Juuti et al., 2013). Because of this, and because of their relative abundance in the atmosphere, aminium sulphates can play a role in the nucleation

and growth of new particles (DePalma et al., 2012; Loukonen et al., 2010; Wang et al., 2010) and in cloud formation (Lavi et al., 2013). However, their physicochemical properties are much less well characterised than their inorganic counterpart, (NH₄)₂SO₄. Some recent work has attempted to fill this gap, reporting the densities (Clegg et al., 2013; Lavi et al., 2013; Qiu and Zhang, 2012), CCN activity, optical properties (Lavi et al., 2013), hygroscopicity and phase transitions of aminium sulphates (Chu et al., 2015; Clegg et al., 2013; Lavi et al., 2013; Qiu and Zhang, 2012; Sauerwein et al., 2015), specifically for

the methyl and ethylaminium sulphates (or subsets of these six compounds). In this work, we report measurements of the hygroscopic behaviour of the three methylaminium sulphates (methylaminium sulphate, MMAS; dimethylaminium sulphate, DMAS; trimethylaminium sulphate, TMAS) and the three ethylaminium sulphates (ethylaminium sulphate, MEAS; diethylaminium sulphate, DEAS; triethylaminium sulphate, TEAS) from water evaporation experiments made using a comparative kinetic Electrodynamic Balance (CK-EDB) (Rovelli et al., 2016). In Section 2 we review the experimental details

and procedures for performing hygroscopic growth measurements using the CK-EDB. In Section 3, we present measurements of the hygroscopic growth of the six aminium sulphate salts, comparing our measurements with literature values and considering the accuracy and reproducibility of measurements with the CK-EDB.

## 2 Experimental

We first describe the CK-EDB technique and the data analysis before describing the method for preparing aqueous starting

solutions of the aminium sulphates.





## 2.1 Hygroscopic Properties from Comparative Kinetics Measurements in a CK-EDB

Electrodynamic trapping of single charged droplets has been used to investigate various properties of confined particles, including optical properties (Barnes et al., 1997), vapour pressures of low volatility compounds (Pope et al., 2010), hygroscopic properties of atmospherically-relevant aqueous solutions (Choi and Chan, 2002; Chu et al., 2015; Peng et al., 2001; Rickards et al., 2013) and of pharmaceutical aerosols (Haddrell et al., 2013; Peng et al., 2000), and evaporation dynamics of aqueous droplets (Heinisch et al., 2009; Shulman et al., 1997; Zobrist et al., 2011). The experimental setup used in this work has been presented in previous publications (Davies, 2014; Davies et al., 2012a, 2012b, 2013; Haddrell et al., 2012; Miles et al., 2012). In particular, the approach used to quantify the hygroscopic properties of single confined particles from comparative kinetics measurements using a CK-EDB has also been previously discussed (Davies et al., 2013) and its application over a wide solution water activity range from 0.5 to >0.99 has been extensively validated (Rovelli et al., 2016).

In a CK-EDB single charged droplets from two solutions with known chemical composition and concentration can be sequentially dispensed on-demand by means of two alternatively operated microdispensers and trapped within the electrodynamic field generated by a set of concentric cylindrical electrodes. This electrode geometry guarantees a stable and tight trapping of droplets within 100 ms from their generation. Droplets are confined in the trapping chamber within a nitrogen stream with controlled mass flow, temperature and relative humidity. Trapped droplets are illuminated by laser light (532 nm) and the resulting elastic scattering light pattern is collected every 0.01 s by means of a CCD camera and used to keep track of changes in size of a single levitated droplet using the geometrical optics approximation (Glantschnig and Chen, 1981). Note that the variations in refractive index due to water evaporation from each droplet are taken into account for an accurate determination of droplet size, as described in a previous publication (Davies et al., 2012b).

The evaporation rate of water from a droplet containing one or more solutes at a certain RH and temperature depends on the hygroscopic properties of the solution. In comparative kinetics experiments, the evaporation rate of probe droplets with known composition and known evaporation kinetics (either pure water or a NaCl solution) is compared to that of sample droplets in order to determine the hygroscopic properties of the solution droplet being studied (Davies et al., 2013; Rovelli et al., 2016). To do so, typical CK-EDB experiment are composed of a series of at least ten alternating probe and sample droplets (Fig. 1a).

The retrieval of the hygroscopic properties from the radius ($a$) vs. time evaporation profiles of a sequence of probe and sample droplets is shown schematically in Fig. 1. Different analyses are carried out for probe (Fig. 1b) and sample droplets (Fig. 1c). Either pure water (as shown in Fig. 1a) or NaCl solution droplets with known initial salt concentration can be used as a probe. The evaporation profiles of the probe droplets are compared with simulations obtained from the evaporation/condensation kinetics model by Kulmala et al. (1993). By doing so, the gas phase RH can be inferred, either from fitting the $a^2$ vs. time evaporation profile of pure water droplets or from the equilibrated radius of the NaCl solution droplets. Both methods, together



with their associated uncertainties and their RH ranges of applicability, have been discussed previously (Davies et al., 2013; Rovelli et al., 2016).

For the analysis of the radius ($a$) vs. time data of sample droplets, the radius data (Fig. 1c, inset panel i) are converted to droplet

mass (inset ii) by using a 3$^{rd}$ order polynomial parameterisation of density as a function of the solute mass fraction, which is discussed below. The mass flux of water leaving the droplet during evaporation (d$m$/d$t$) is then calculated and, knowing the gas phase RH from the probe droplet analysis, Kulmala's equations for evaporation kinetics are applied to calculate the temporal variation of water activity ($a_w$) in the droplet (inset iii). In addition, knowing the initial concentration and size of the droplet at generation, the radius of the dry particle ($a_{dry}$) can be estimated and a radial growth factor ($GF_r = a / a_{dry}$) can be

calculated for each of the measured radii (inset iv). Results from inset panels (iii) and (iv) are then combined and typical $GF_r$ vs $a_w$ growth curves are obtained. The key thermodynamic quantities that describe the hygroscopic properties of the tested solution (moles of water per mole of solute in solution, $n_{water}/n_{solute}$; osmotic coefficients, $\phi_{st}$) can be calculated if the densities of the aqueous solutions are also known (see below). Details of the treatment of the experimental uncertainties and their influence on each of the computed quantities are described in the Supplementary Information (Table S1). When error bars are

not shown in the figures in the following sections, the reader can assume that they are smaller than the size of the corresponding data point.

A previous study (Cai et al., 2016) showed that the molar refraction mixing rule, together with a 3$^{rd}$ order polynomial parameterisation of density as a function of the square-rooted mass fraction of solute ($mfs$), represent the best approach to

predicting refractive indices ($m$) and densities ($\rho$) of solutions of organic compounds for which bulk data of such quantities is available for solute mass fractions up to at least 0.4. In this work the densities of at least ten solutions with different concentrations for each aminium sulphate were measured with a density meter (Densito 30PX, Mettler Toledo, accuracy of ±0.001 g cm$^{-3}$, calibrated with pure water before each use). Densities were measured at ambient temperature, which varies in the laboratory between 293 K and 295 K; temperatures were always registered together with the measured density values. In

addition, refractive indices of the same solutions were measured at 589 nm by means of a refractometer (Palm Abbe II, Misco, precision of ±0.0001, calibration with pure water before each use). The measured density and $m$ values for each aminium sulphate solution are provided in the Supplementary Information (Table S2), together with the 3$^{rd}$ order polynomial and the molar refraction mixing rule fittings for each compound (Table S3 and Fig. S1). A brief description of the molar refraction mixing rule application is also provided in the Supplementary Information. These data have been presented previously and

discussed by Cai et al. (2016) along with measurements from a large number of organic aqueous solutions.



## 2.2 Preparation of the Solutions

Aminium sulphate stock solutions were prepared by the neutralization of solutions of each of the six amines with aqueous sulphuric acid. The commercial amines stock solutions (Sigma Aldrich, MMA, ~ 40 wt%; DMA, ~ 40 wt%; TMA, ~ 45 wt%; MEA, ~ 66.0-72.0 wt%; DEA ≥ 99.5 wt%; TEA ≥ 99 wt%) were titrated with standardised HCl (1 M, SLS) to determine their mass concentrations accurately. Three repetitions were performed for each titration and the pH was measured throughout by means of a pH-meter (HI 8314, Hanna Instruments), which was calibrated with standard pH 7 and pH 4 solutions. Before the titration, the amine stock solutions were always diluted down to 1-5 wt% in order to minimize the heat generated by the neutralisation reaction and to minimise volatilisation of the amine. In addition, the HCl (and later the $H_2SO_4$) used for the standardisation of the commercial amine solutions were titrated with $Na_2CO_3$ (≥ 99.5%, Alfa Aesar), which was first dried at 225°C for 3 h before weighing to make sure that no water was adsorbed on it. Three repetitions were performed in these cases.

For the preparation of the aminium sulphates stock solutions, stoichiometric amounts of the standardised $H_2SO_4$ and amine solutions were mixed, with an initial concentration of both solutions around 40 wt%. The pH of the solution mixture was monitored for the whole duration of the reaction to ensure that all of the amine in the solution had reacted. The concentrated stock solutions of the salts that result from this procedure were subsequently diluted down to a mass fraction of ~0.05, in order to obtain a suitable starting concentration for the CK-EDB comparative kinetics measurements. During both the titration of the amine stock solution with HCl and the preparation of the aminium sulphates solutions with $H_2SO_4$, the amine solution was kept in an ice bath and the addition of the acid was performed slowly and dropwise, in order to dissipate the heat generated by the neutralisation reaction and to avoid any possible amine volatilization. Phase separation was observed when titrating the TEA commercial solution and its concentration was determined to be 81.6 wt% (0.52 wt% standard deviation over 3 repetitions), which is considerably lower than the ≥ 99 wt% concentration value given by the manufacturer. This is possibly due to the much lower solubility of TEA (0.7 mol kg$^{-1}$) than the other amines (Ge et al., 2011b), leading to incomplete solvation of the amine in water and an inaccurate measurement of pH during the titration. Thus, we instead assumed that the commercial TEA solution was a 99 wt% concentration and note that the results for the TEA system should be interpreted with some caution. This experimental procedure ensured that the concentrations of the reagents were well known and, consequently, that the concentrations of the stock solutions prepared for aminium sulphates were similarly well known. The uncertainties in the commercial solution concentrations of the reagents, as determined from the repeated titrations, were taken into account for the calculation of the overall experimental uncertainties indicated in the Supplementary Information (Table S1).

To validate the procedure for solution preparation described above, the steps were carried out for the preparation of $(NH_4)_2SO_4$ from the reaction of ammonia and sulphuric acid. The resulting salt solution was then used in comparative kinetics measurements and the hygroscopicity of the ammonium sulphate from reaction was compared with calculations from the Extended Aerosol Inorganics Model (E-AIM) (Wexler and Clegg, 2002). Figure 2 shows the hygroscopic properties of





$(NH_4)_2SO_4$ droplets prepared in this way from the direct reaction of ammonia and sulphuric acid, reporting values of $n_{water}/n_{solute}$, vs. $a_w$ (Panel (a)) and osmotic coefficients ($\phi_{st}$) plotted against the square-root of the sulphate molality ($m(SO_4^{2-})^{0.5}$, Panel (b)). Osmotic coefficients are useful parameters to represent the deviation of a solution from an ideal behaviour and they are defined as indicated in Eq. (2):

$$\phi_{st} = -\frac{\ln(a_w)}{M_w 3m/1000} \tag{2}$$

where $M_w$ is the molecular weight of water, $m$ is the molality of the solute and 3 is the stoichiometric number of ions in the salt. The value of $\phi_{st}$ tends to 1.0 in the limit of an infinitely dilute solution, in accordance with the Debye-Hückel limiting law (Robinson and Stokes, 1970).

In a previous publication (Rovelli et al., 2016) we showed that it is possible to achieve very good agreement with predictions from the E-AIM model for well-characterised inorganic compounds with the CK-EDB experimental technique within an uncertainty in $a_w$ of ±0.002. The plots in Fig. 2 show the averaged data obtained from two datasets of ten droplets of aqueous $(NH_4)_2SO_4$ (black and open circles). In Fig. 2b the effect of a ±0.002 error on $a_w$ on the modelled osmotic coefficients values is shown with dashed lines. The osmotic coefficients agree well with calculations from the E-AIM model and lie within the envelope associated with this previous estimate of typical experimental uncertainty for the CK-EDB technique. This demonstrates that the volatility of ammonia is not a significant problem when the neutralisation reaction with $H_2SO_4$ is performed. In addition, since the vapour pressure of ammonia (1956 kPa at 298.15 K) (Lide, 2005) is even higher than the vapour pressure of the most volatile of the six considered amines (methylamine, 336 kPa at 298.15 K) (Ge et al., 2011b), it is likely that there is also no evaporative loss of amines from solution during the preparation of the aminium sulphate solutions. This result confirms that the estimated concentrations of the $(NH_4)_2SO_4$ solutions prepared from $NH_3$ and $H_2SO_4$ are accurate, and that the preparation method is reliable for both $(NH_4)_2SO_4$ and all the six aminium sulphates.

## 3 Results and Discussion

We first report our measurements of the hygroscopic response of the sequence of six aminium salts before comparing our results with previous studies and assessing the accuracy and reproducibility of our data.

## 3.2 Hygroscopic Properties of Aminium Sulphate Droplets

The hygroscopic properties of the series of six aminium sulphates were characterised by means of comparative kinetics measurements and using the density and refractive index parameterisations discussed for the CK-EDB data treatment, as described in Sect. 2.1. First, the radial growth curves are shown in Fig. 3a. Considering the compound with the lowest molecular weight first, the hygroscopic behaviour of MMAS is the most similar to ammonium sulphate, in terms of $GF_r$.





Continuing in the methylaminium sulphates series, a slight decrease in $GF_r$ is observed in the high water activity region ($a_w$>0.8) for DMAS and TMAS, while at lower water activities, the hygroscopic properties of these compounds converge to $(NH_4)_2SO_4$ within the uncertainties of the measurements as the amount of water in the particles decreases. With respect to the ethylaminium sulphates series, a more evident decreasing trend in the radial growth curve is observed with increasing number

of C atoms in the cation (MEAS > DEAS > TEAS), once again especially in the upper part of the curves. If the mono-, di- and tri- pairs within the two different series are compared, the methyl compound always presents higher values of radial growth factor than its equivalent in the ethylaminium sulphates series.

For the calculation of the dry radius reference state in the denominator of $GF_r$, the pure melt density is used (Sect. 2.1); if the

10 pure solid density values were known and used, one could expect that the calculated radial growth curve would be slightly higher, a consequence of the solid densities having higher values than the melt density (Clegg et al., 2013). A comparison of the pure melt densities ($\rho_{melt}$) from our work and from Clegg et al. (2013) and solid densities ($\rho_{solid}$) estimated by Qiu and Zhang (2012) (Qiu and Zhang, 2012) is provided in Table S4 in the Supplementary Information, with our data previously published in Cai et al. (2016). As an example, if a hypothetical increase of 5% from $\rho_{melt}$ to $\rho_{solid}$ as a rough estimate is considered for

DMAS, the obtained $GF_r$ curve would increase by less than 1% (less than 0.01 in $GF_r$); this would not affect the trends shown in Fig. 3a, although it would marginally change the relative position of the aminium sulphates curves to that of ammonium sulphate.

The same datasets shown in Fig. 3a have been converted to $n_{water}/n_{solute}$ vs. $a_w$ and are shown in Fig. 3b. Note that the $y$-axis is

20 displayed on a logarithmic scale for clarity. Different trends can be recognised: the primary factor influencing how many moles of water are absorbed per mole of solute is the number of alkyl groups in the cation and not the length of the carbon chains (i.e. mono- vs. di- vs. tri-methyl or ethyl). In fact, when the $n_{water}/n_{solute}$ curves (Fig. 3b) are compared with the hygroscopic behaviour plotted as $GF_r$ (Fig. 3a), more significant differences are apparent in Fig. 3b within each series (e.g. among MMAS, DMAS and TMAS) rather than between analogous compounds belonging to the two series (e.g. between TMAS and TEAS),

even though the ethylaminium sulphate in a pair always has slightly higher values of $n_{water}/n_{solute}$ (i.e. MMAS ≈ MEAS < DMAS ≈ DEAS < TMAS ≈ TEAS).

The seemingly different trends found for the hygroscopic properties of aminium sulphates when reported either as $GF_r$ or $n_{water}/n_{solute}$ as a function of $a_w$ originate in the different physicochemical properties that each quantity is dependent on. When

$GF_r$ is calculated, the density and molecular weight of each compound are central to its determination. For example, if particles with a dry radius of 1 μm for each of the six aminium sulphates are considered, the moles of salt in each droplet decreases with increase in molecular weight and decrease in density, as shown in Fig. 4. Therefore, if hygroscopicity is represented by a radial growth factor, $GF_r$ is determined not only by the ability of a compound to absorb water at a certain relative humidity, but also





by the number of solute molecules present in the droplet itself. For this reason, a decreasing hygroscopic growth trend with increasing molecular weight of the aminium sulphates is apparent in Fig. 3a for $GF_r$.

Growth factor curves are widely used in the aerosol literature, especially when the optical properties and radiative forcing or the climatic effects of atmospheric aerosols are investigated, because these properties largely depend on particle size. The quantification of the hygroscopic properties of a compound in terms of $GF_r$ is therefore useful for such applications and is reported in this work. However, $n_{water}/n_{solute}$ vs. $a_w$ curves allow the thermodynamic characterisation of the water uptake of a substance and the decoupling of its hygroscopic properties from its density and molecular weight, with the water uptake 'normalised' to the moles of solute in the particle. This representation – effectively water activity as a function of concentration – is mostly used when focusing on chemical reactivity or on the thermodynamic properties of compounds in the aerosol phase. The increasing trend apparent in Fig. 3b for $n_{water}/n_{solute}$ curves with increasing number of carbon atoms in the cation is related to the size of the cation itself: the bigger the cation, the larger the number of water molecules required to solvate the cation at a particular water activity. It is perhaps worth noting here the high level of accuracy in retrieving hygroscopic growth by this method and the opportunity it provides to resolve such fine trends in growth factor.

## 3.2 Comparisons with Previous Studies

As described in the Introduction, there have been some recent reports of the physicochemical properties of aminium sulphate aerosols and aqueous solutions, motivated by the increasing understanding of their role in atmospheric processes (Bzdek et al., 2010; Ge et al., 2011a; Lavi et al., 2015; Liu et al., 2012; Qiu and Zhang, 2013). With respect to the determination of their hygroscopic properties, the approach presented here differs from the earlier reports in either the preparation method of the solutions or in the experimental measurement technique. We now compare our measurements using a CK-EDB with the results of these previous studies.

Qiu and Zhang (2012) were the first to measure diameter growth curves for these compounds (excluding MEAS) by means of an HTDMA (Hygroscopicity Tandem Differential Mobility Analyser). They inferred hygroscopic growth factors from the ratio of the mobility diameters measured at a variable RH (up to 90%) and at an RH of ~12%. A monotonic increase in the particle size with RH for each compound was observed and no deliquescence/efflorescence behaviour was observed. If their growth curves are compared with the $GF_r$ shown in Fig. 3a (Fig. S2 in Supplementary Information), a systematically smaller growth factor is reported from the HTDMA measurements. The largest deviations are for TEAS (-16% on average in the $a_w$ range where the two datasets overlap, from ~ 0.5 to 0.9) and for TMAS (-13%), a deviation of -10% is observed for DEAS and DMAS and of -7% for MMAS. These discrepancies are most likely due to the presence of some residual water at the conditions at which the reference diameter was measured in the HTDMA experiments (RH ~12%), which would result in an overestimated reference 'dry' size and in underestimated growth factor values. This explanation is supported by the studies of



Chan and Chan (2012) who reported the presence of water for some aminium sulphates even at an RH of ∼3%. In addition, the possible volatilisation of the amine during the drying step in the HTDMA would alter the chemical composition of the particles (Chan and Chan, 2012, 2013), not only artificially reducing the apparent dry size recorded but shifting the dry particle towards an aminium bisulphate composition (1:1 molar ratio of sulphuric acid to amine), which has been shown to be less

hygroscopic than its sulphate counterpart (Sauerwein et al., 2015).

Consistent with the above discussion, the two compounds with the biggest deviation between the $GF_r$ values reported here and by Qiu and Zhang (TMAS and TEAS) are those that were found to be affected by the largest evaporative losses of amine by Chan and Chan (2012). They reported studies of ammonium displacement reactions by alkylamines by levitating single

droplets in an EDB and evaluating the changes in the recorded Raman spectra during an experiment. After TEAS solution droplets were levitated at RH<3% for more than 5 h, TEAS was found to have converted to TEA bisulphate almost completely, indicating that half of the amine in the initial solution droplet had evaporated. Similar behaviour was observed for TMAS, while DMAS and DEAS showed a smaller degree of evaporation of the amine (∼25% and ∼5%, respectively); MMAS and MEAS did not show any relevant evaporation over the experimental timescales. These results support the hypothesis of

possible evaporation of the amine from the drying solution droplets during HTDMA experiments, and at the same time represent an interesting comparison to evaluate the timescales over which the evaporative loss of amine is significant. The hygroscopic growth measurement of each droplet of aminium sulphate solution in CK-EDB studies extends over 30 s at the longest: this allows the effective decoupling of the fast water evaporation and the slow amine evaporation, since these two processes occur over different timescales. In addition, the hygroscopicity measurements in the present work were carried out

at RHs in excess of 50% at all times; for this reason, the evaporation of the amines are necessarily smaller than those measured by Chan and Chan (2012).

Clegg et al. (2013) have converted the size growth curves measured by Qiu and Zhang to $n_{water}/n_{solute}$ vs. $a_w$. The growth curves that resulted did not show any discernible trends in the hygroscopicity of aminium sulphates. Indeed, the results were

essentially similar to the water uptake of ammonium sulphate in the lower $a_w$ range. At higher $a_w$ above 0.7, the scatter in the data limited the identification of any apparent trend apart from the suggestion that the methylaminium sulphates absorb fewer moles of water per moles of solute than the compounds in the ethyl series. Fig. 5 shows the large differences between the water uptake determined from the results of Qiu and Zhang, and that from the work in this study. Because of the uncertainties associated with the HTDMA $GF_r$ values, it is hard to draw conclusions the comparison with our results.

Sauerwein et al. (2015) have recently reported bulk water activity measurements with an activity meter (Aqua lab Series 3TE) for various amine-to-sulphate ratios and over a concentration range of dissolved electrolyte up to 9 mol kg⁻¹ at 25°C. (Note that TEAS was not considered in their study.) Using a bulk measurement technique for the determination of hygroscopic properties of aminium sulphates has the advantage of limiting possible evaporative losses of the amine during the experiments,





especially compared to accumulation-mode aerosol measurements with an HTDMA for which timescales of evaporative loss/equilibration are assumed very short. As a first comparison, the hygroscopic growth curves previously shown in Fig. 3b in terms of $n_{water}/n_{solute}$ vs. $a_w$ are compared with results from Sauerwein et al. (2015) (Fig. 6a in their paper) in Fig. 5. The hygroscopic growth curves determined from the two different experimental methods differ marginally, but these discrepancies

are considerably smaller than the differences between the results of both studies – CK-EDB and bulk water activity measurements – and HTDMA data from Clegg et al. (2013).

A comparison of the two datasets represented in terms of osmotic coefficients (Eq. (2)) can provide further insight, see Fig. 6. Notably, the measurements reported here extend to both more dilute and concentrated solutions and include measurements for

TEAS. If the errors associated with each dataset are considered, the two sets of measurements overlap over some of the range in $m(SO_4^{2-})^{0.5}$, especially in the low molality limit (high water activity region) for MMAS, TMAS, MEAS and DEAS ($m(SO_4^{2-})^{0.5} < 1$, approximately), but also in the region $m(SO_4^{2-})^{0.5} > 2$ for DMAS and TMAS. A comparison with the uncertainty in the osmotic coefficients that arises from an uncertainty of $\pm0.002$ in water activity (same as in Fig. 2b) indicates that the observed discrepancies are larger in magnitude than this typical experimental uncertainty of our technique, except for very low sulphate

molality values (below ~0.8). However, despite the discrepancies that do exist, results from the two different experimental methods show the same qualitative hygroscopicity trends over the methyl and ethylaminium sulphates series. For further comparison, the results of Clegg et al. (2013) (from the HTDMA data) were converted to stoichiometric coefficients and are plotted in Fig. 6. These values are broadly comparable for all of the compounds, and close to those for ammonium sulphate as noted earlier. They do not agree with either the results of the experiments reported here, or the bulk measurements of Sauerwein

et al. (2015).

We now consider the differences that exist between the data of Sauerwein et al. (2015) and our measurements. It is worth stating here that the evaporation kinetics measurements at the core of the CK-EDB approach have been validated in a previous publication (Rovelli et al., 2016) by means of the determination of the hygroscopic properties of well-characterised inorganic

compounds and their mixtures. In addition, we have performed sensitivity tests to evaluate possible effects of random experimental errors associated with the proposed experimental method. The reproducibility of our measurements and the uncertainties associated with the treatment of densities are discussed below in Sect. 3.3 and 3.4, where they are shown not to affect significantly the determined hygroscopic properties of a compound.

If the differences between our measurements and the results of Sauerwein et al. (2015) were caused by partial volatilisation of the amine from the droplets evaporating in the CK-EDB, the observed bias between the two datasets would be reversed, i.e. an underestimation of $n_{water}/n_{solute}$ and of osmotic coefficients would be expected, similar to what has been discussed in the case of the HTDMA size-based measurements of Qiu and Zhang (2012) when converted to a molar basis by Clegg et al. (2013). Another potential significant source of error could be inaccuracies in the starting concentrations of the aminium sulphate



solutions. In this respect, the solution preparation and measurement method described in Sect. 2.2 has been validated through the reaction of ammonia and sulphuric acid and the measurements of the hygroscopic properties of the obtained $(NH_4)_2SO_4$. If the volatilisation of $NH_3$ during the neutralisation reaction was not an issue in preparing the solution of $(NH_4)_2SO_4$ it is unlikely to present a problem for the alkylamine solutions, with all alkylamines characterised by lower vapour pressures than ammonia

5 (Ge et al., 2011b). Furthermore, in order to achieve complete agreement with the data from Sauerwein et al. (2015) the aminium sulphates solutions should be 5-10% more concentrated than calculated from the titration of the commercial stock solutions and from the amount of the reagents. Not only does this percentage seem unrealistically high, but the solute concentrations would be overestimated and not underestimated if any volatilisation of the amine occurred during the preparation of the solutions.

Sauerwein et al. (2015) provide a framework for the estimation of the water content in mixtures (in this case ternary mixtures of $H_2O$-$H_2SO_4$-amine) at any molar ratio of solutes. This allows a calculation of the water content (in terms of kg of water per mole of solute, or molality) and the stoichiometric osmotic coefficients (Eq. (2)) for the relevant aminium:sulphate ratio of 1:1 (bisulphate) and 2:1 (sulphate). However, it should be noted that unlike the study presented here, measurements were not explicitly made for the 2:1 molar ratio aqueous aminium sulphate solution. Instead, Sauerwein et al. used a modified

Zdanovskii- Stokes-Robinson expression (Eq. (3) of Sauerwein et al.) to represent the water uptake of the different mixtures of aqueous aminium sulphates and $H_2SO_4$; these fits are shown in Fig. 7 along with the actual measured points. $x_{salt}$ can be considered as the degree of neutralisation of sulphuric acid. $x_{salt}$=0 corresponds to pure $H_2SO_4$ and the water content in this case is well-known (Clegg and Brimblecombe, 1995); the ZSR fit is therefore constrained to this value. A value at $x_{salt}$=1 corresponds to the completely neutralised aminium sulphate. The results of the fit were used to estimate the water uptake of

each solution of aqueous aminium sulphate (i.e. the 2:1 molar ratio at $x_{salt}$=1) and it is these values that are shown in Fig. 6. In order to compare our data with that of Sauerwein et al., values at the same water activities must be compared (0.8 and 0.925 are chosen here) and a linear interpolation between actual measurement points is required to achieve this. Normally, the data from CK-EDB measurement are calculated with $a_w$ steps of 0.01 for $a_w > 0.8$ where hygroscopic growth curves are the steepest,

with a spacing of 0.02 for the water activity range ~0.65 - 0.8 and of 0.03 for $a_w < 0.65$. As in Sauerwein et al. (2015), the uncertainty of the interpolated points is set as the largest uncertainty among the experimental points used for the interpolation. The discrepancies between the estimations from bulk and CK-EDB measurements shown in Fig. 5 are apparent in Fig. 7 too at $x_{salt}$=1: the amount of water associated with each mole of aminium sulphate is systematically higher from the CK-EDB measurements when compared with the bulk measurements at both $a_w$ with the discrepancy higher at the higher water activity.

In view of the differences between our measurements at $x_{salt} = 1.0$ and the extrapolations from the fits of Sauerwein et al., we include a second fit to the ZSR model used previously but including our own measurements of the water uptake. The fit equations and parameters are provided in Supplementary Information (Table S6). As well as the fitted values more closely reflecting our measured values for the pure aminium sulphates at $x_{salt}$ equal to unity, the revised fit does not significantly





worsen the fit to the measurements of Sauerwein et al. for the acidified mixtures when $x_{salt} < 0.6$). We therefore suggest that the discrepancies seen in terms of $n_{water}/n_{solute}$ (Fig. 5), osmotic coefficients (Fig. 6) and kg of water per mole of solute in Fig. 7 can be attributed, in part, to the uncertainties associated with the ZSR fits and interpolation necessary to estimate the water uptake of the pure aqueous aminium sulphates from the results of Sauerwein et al. (2015). Certainly there is no physical reason

why a simple relation such as the modified ZSR equation (Eq. (3) of Sauerwein et al.) should exactly represent the relationship between water activity and chemical composition in such systems, especially considering the large and unquantified influence of the sulphate-bisulphate equilibrium on the thermodynamic properties of the solutions.

A possible constant error in measured water activity could be the origin of the differences, but this seems unlikely. Such an

offset in water activity results in a large change in the osmotic coefficient at the lowest concentrations for which measurements were made, but a much smaller change at the highest concentrations. This is not consistent with the differences in $\phi_{st}$ shown in Fig. 6. As a final consideration, we now explore the reproducibility of the CK-EDB measurements and the sensitivity of the retrieved hygroscopicity to the chosen treatment for solution density.

## 3.3 Reproducibility of Measurements

In order to evaluate the reproducibility of the solution preparation method (Sect. 2.2) and of the retrieval of hygroscopic properties of a compound from the CK-EDB measurement, data obtained from three different datasets of evaporating DMAS solution droplets are compared in Fig. 8. The variation in $n_{water}/n_{solute}$ with $a_w$ (Panel (a)) and the change in osmotic coefficient with square root of sulphate mass fraction (Panel (b)) are calculated from three different data sets, each arising from

measurements with 10 droplets and prepared from different starting stock solution of diethylaminium sulphate, obtained separately one from the other by mixing DEA and $H_2SO_4$ in different days. Data from Sauerwein et al. (2015) (open circles) and calculations for ammonium sulphate (E-AIM model, line) are shown for comparison. Measurements were made with droplet evaporation into a gas phase RH of ~80%. The reproducibility of the data is very satisfactory, giving further evidence that the experimental method is reliable. In addition, this consistency in experimental reprodubility strongly suggests that the

discrepancies with the data of Sauerwein et al. (2015) do not originate from random errors associated with the CK-EDB experiments.

## 3.4 Sensitivity to Parameterization of Solution Density

A knowledge of the solution density as a function of solute concentration is needed to the process CK-EDB evaporation radius

profiles (Sect. 2.1). The effect of uncertainties in the density parameterization used is here evaluated for DMAS, in order to estimate how potential errors in the density measurements and uncertainties in the extrapolation of $\rho_{melt}$ at $mfs$ equal to unity could affect the hygroscopicity data retrieved from comparative kinetics measurements. In Fig. 9, the densities of bulk DMAS





solutions measured in this work are shown together with their 3$^{rd}$ order polynomial fit, which is the parameterization used in all the calculations for this compound in this work (Table 1). As shown in a previous work (Cai et al., 2016), a ±2% uncertainty is typical for the estimated melt density ($mfs$ = 1.0) predicted from with a 3$^{rd}$ order polynomial fit of measured solution densities (Sect. 2.1), while the error is decreases with $mfs^{0.5}$ for the rest of the curve as shown in Fig. 9. More generally, the aminium

sulphates are very soluble compounds and it was possible to directly measure the density of their bulk solutions over a wide range of solute mass fractions (up to $mfs^{0.5}$ values of 0.67-0.85, depending on the compound, see Table S2). Consequently, the $mfs^{0.5}$ range over which the extrapolation for the calculation of $\rho_{melt}$ is needed is small and an uncertainty of ±2% (Fig. 9) is large for the evaluation of its effects. The two 3$^{rd}$ order polynomials (called 'Error+' and 'Error-' in Table 1, respectively) were applied together with the molar refraction mixing rule to explore the sensitivity of the data analysis of the original DMAS

evaporation kinetics datasets to the representation of the solution density. Density data from Clegg et al. (2013) are also shown in Fig. 9 for comparison. A small discrepancy between their measurements and the data presented in this work is observed, not only for DMAS but also for the other 4 aminium sulphates they considered; but no clear pattern to these differences. Figure S3 shows a further comparison of density data from this work and from Clegg et al. (2013) converted in terms of apparent molar volumes; both measured and fitted apparent molar values are provided and the coefficient of the fitted equations can be

found in Table S5. They may be attributable to the different preparation procedures of the aminium sulphates solutions and to the different experimental techniques for the measurement of densities.

The hygroscopic properties of DMAS obtained from the treatment of the evaporation kinetics data with the 'original' density parameterization and with the upper and lower bounds on the density treatment are shown in Fig. 10. Data from Sauerwein et

al. (2015) and calculations for ammonium sulphate (E-AIM model) are plotted for comparison. When the hygroscopic properties of DMAS are represented either in terms of the dependence of either $n_{water}/n_{solute}$ or $GF_r$ on water activity (Panels (a) and (b)), the three curves deriving from the three different density treatments are virtually undistinguishable; thus the uncertainty in the applied density parameterisation does not significantly alter the analysis. In the case of the osmotic coefficients plot (Panel (c)) some very slight deviations between the three treatments can be distinguished at the two extremes

of the plot. With respect to the low sulphate molality region, these small differences are due to the fact that a small variation in the sulphate molality results in more significant variations in the osmotic coefficients, because $m_{st}$ appears in the denominator in the osmotic coefficient expression (Eq. (2)). If the high sulphate molality region is considered, the variations among the three curves are more significant because the simulated error on the density parameterisation is larger for more concentrated solutions (i.e. the size of the grey envelope in Fig. 9).

It is clear that the hygroscopic properties of aminium sulphates determined from CK-EDB measurements are relatively insensitive to reasonable variations in the extrapolated $\rho_{melt}$ value (±2%) and to the applied density parameterisation. In addition, the variations introduced by different density parameterisations are very small compared to the differences from the



results of Sauerwein et al., who obtain lower values of the hygroscopicity and osmotic coefficient. We conclude that these differences cannot be caused by inaccuracies in the approaches for treating variations of the density of evaporating droplets.

## 4 Summary and Conclusions

Quantifying the hygroscopic properties of aminium sulphates is important for an accurate understanding and modelling of some atmospheric processes in which they are involved (e.g. formation and growth of new particles, activation of cloud droplets). In order to measure their hygroscopic properties with a CK-EDB, the dependence of aminium sulphates solution densities and refractive indices on mass fraction of solute are reported. The experimental technique together with the application of the molar refractive mixing rule and a 3$^{rd}$ order polynomial parameterisation of density for the representation of

the refractive indices and densities of solutions with variable solute mass fractions were presented and validated in previous works (Cai et al., 2016; Davies et al., 2013; Rovelli et al., 2016).

The procedure for preparation of the aminium sulphates stock solutions was validated by using it for the preparation of ammonium sulphate solutions from the direct reaction of ammonia and sulphuric acid. The hygroscopic properties of the

obtained $(NH_4)_2SO_4$ solutions retrieved from CK-EDB measurements were in very good agreement with calculations from E-AIM model and we demonstrated that the uncertainty of such measured values was comparable with a typical uncertainty of ±0.002 on $a_w$, which was previously shown to be typical for CK-EDB measurements (Rovelli et al., 2016). This result demonstrated that the used preparation procedure is robust and reliable.

The experimental results for the aminium sulphates were compared with the few studies available in the literature and the observed discrepancies were discussed in the light of the different experimental approaches. Measurements from our new approach provide a level of accuracy that reveals clearly the fine variations in hygroscopic growth that occur with molecular structure and substitution, and avoids the additional complexity of volatilisation of semi-volatile components during hygroscopic growth with measurements complete in a matter of a few seconds. The largest discrepancies with previous data

were found when comparing the CK-EDB results with H-TDMA measurements (Clegg et al., 2013; Qiu and Zhang, 2012). These differences can be attributed to a possible overestimation of the dry size of particles due to residual water in the reference dry state in HTDMA measurements and/or to a shift of the chemical composition of particles towards the bisulphate composition because of partial volatilization of the amines from solution. More comparability was found with the bulk water activity measurements by Sauerwein et al. (Sauerwein et al., 2015). The main differences in approaches are that: we perform

aerosol measurements that cover a wider range in water activity as compared with the bulk measurements of Sauerwein et al. (2015); and we provide direct measurement at amine-to-sulphate ratios of exactly 2:1, whereas Sauerwein et al. (2015) performed a ZSR fitting on data from solutions with variable amine-to-sulphates ratios and extrapolated water content for the exact 2:1 ratio. These new CK-EDB measurements suggest a higher level of hygroscopic growth for the aminium sulphates



than previously reported by Sauerwein and co-workers when inferred from measurements over a range of amine-to-sulphates ratios; we have provided a refined parameterisation for all compositions.

As a final remark, it is worth noting that the characterization of the hygroscopic properties of aminium sulphates up to $a_w$ of 0.99 was possible with the CK-EDB technique. The other literature approaches that were discussed in this Section were able to cover larger $a_w$ ranges (down to 0.1) but none of them could be applied to obtain any data for $a_w > 0.9$. Thus, the comparative kinetics measurements in a CK-EDB provide a powerful tool for investigating a water activity region that is otherwise hard to characterise with such accuracy, but which is of great importance for the understanding of the activity of aerosol particles as cloud condensation nuclei (Wex et al., 2009).

Aminium sulphates are the first class of mixed inorganic-organic aerosol systems to be investigated by means of the CK-EDB comparative kinetics technique, described in Rovelli et al. (2016), over a wide range of water activities. Therefore, besides the atmospheric relevance of these compounds, this study also provides a deeper understanding of the possible effects caused by random errors in the experimental procedure, and by uncertainties on the representation of the density of a compound. We demonstrated that the CK-EDB measurements are characterised by a very good level of reproducibility and that a typical ±2% uncertainty on the value of the extrapolated melt density only marginally affect the measured hygroscopic properties.

**Acknowledgments**

REHM, JPR, and SLC acknowledge support from the Natural Environment Research Council through grant NE/N006801/1. G.R. acknowledges the Italian Ministry of Education for the award of a PhD studentship.

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





**Tables**

| | 3rd order polynomial density coefficients | | | | Melt | Melt $\rho$ / |
| | $a$ | $b$ | $c$ | $d$ | R.I. | g cm$^{-3}$ |
|---|---|---|---|---|---|---|
| **DMAS** | 0.99847 | -2.55·10$^{-3}$ | 0.34103 | -0.05191 | 1.4725 | 1.2850 |
| **Error+** | 0.99847 | 0.017256 | 0.342053 | -0.04702 | 1.4665 | 1.3108 |
| **Error-** | 0.99847 | -0.02231 | 0.339909 | -0.05675 | 1.4783 | 1.2593 |

5     **Table 1: Parameters of the 3rd order polynomial parameterisation for density and for the molar mixing rule (Sect. 2.1). DMAS is the original fitting of experimental data (black circles in Fig. 9), while 'Error+' and 'Error-' are calculated supposing a ±2% error on the extrapolated $\rho_{melt}$, as described in the text.**



**Figures**

**Figure 1: Schematics representing the retrieval of hygroscopic growth curves from comparative kinetics experiments in a CK-EDB.**





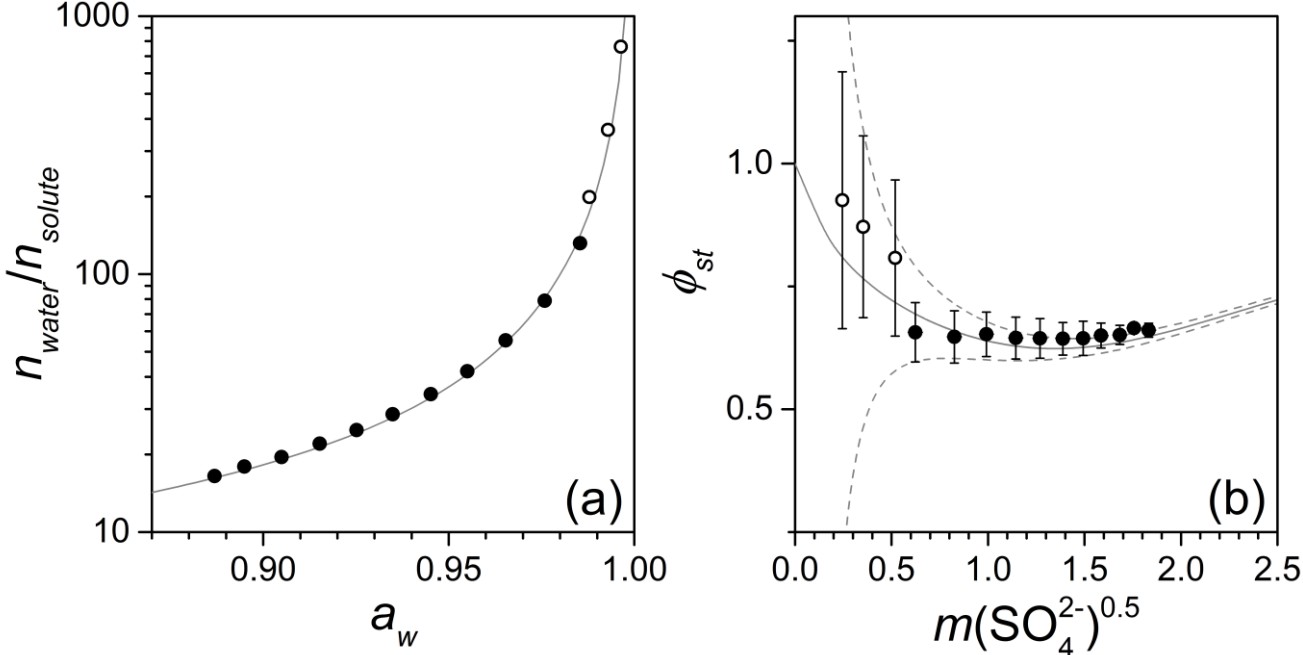

**Figure 2: Measured $n_{water}/n_{solute}$ vs. $a_w$ (Panel (a)) and osmotic coefficients ($\phi_{st}$) vs. the square root of sulphate molality ($m(SO_4^{2-})^{0.5}$) (Panel (b)) of ammonium sulphate solution obtained from the reaction between NH₃ and H₂SO₄. Symbols: black dots – 0.03 *mfs* of (NH₄)₂SO₄ in the initial solution, 88.5% RH in the gas phase; open circles – 0.004 *mfs* of (NH₄)₂SO₄ in the initial solution, 90% RH in the gas phase; solid lines – calculations from E-AIM model; dashed lines – uncertainty on the osmotic coefficients corresponding to an error in $a_w$ of ±0.002.**





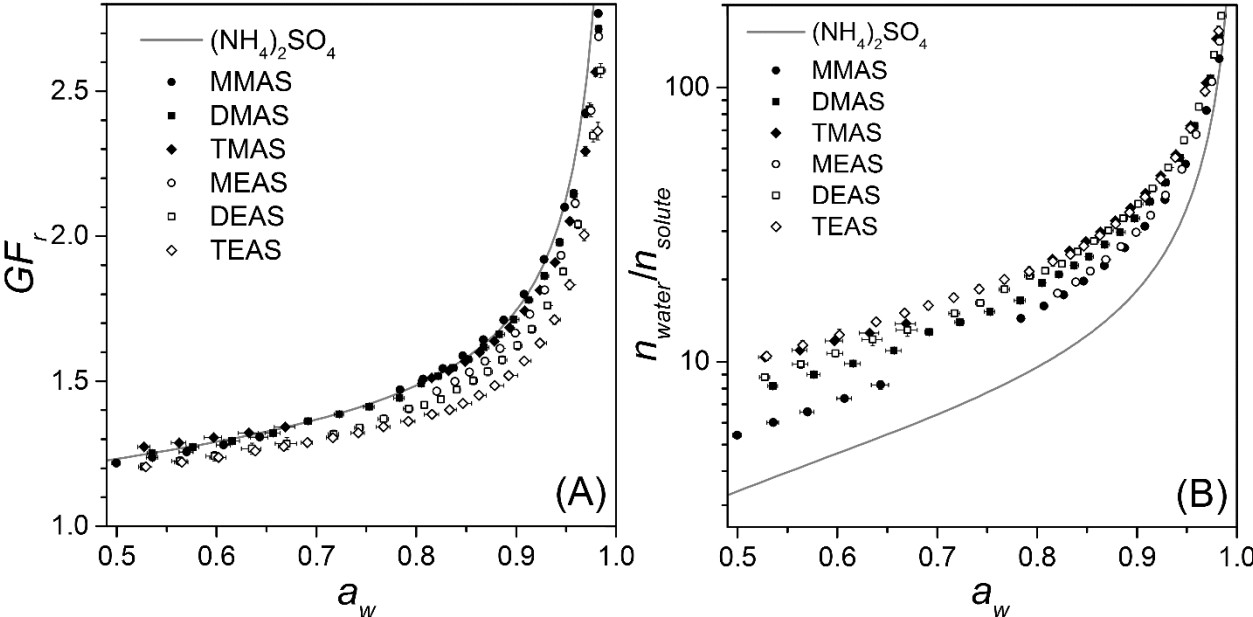

**Figure 3: Aminium sulphates $GF_r$ vs. $a_w$ (Panel (a)) and $n_{water}/n_{solute}$ vs. $a_w$ (Panel (b)) hygroscopic growth plots from CK-EDB experiments. Symbols: ● – MMAS; ■ – DMAS; ♦ – TMAS; ○ – MEAS; □ – DEAS; ◇ – TEAS; line, E-AIM model calculation for (NH₄)₂SO₄.**



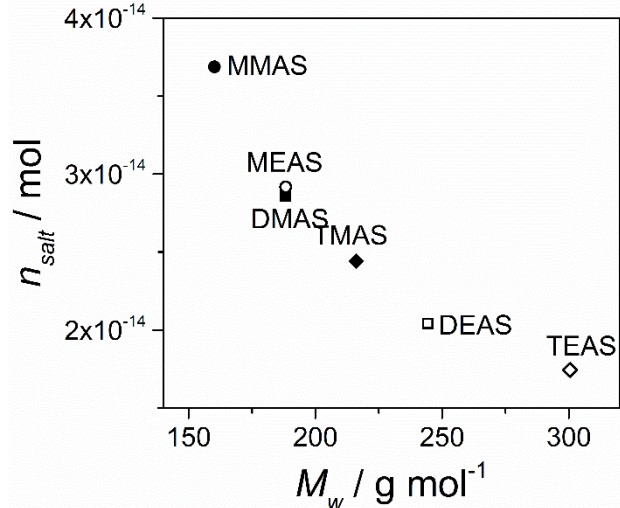

**Figure 4: Moles of salt ($n_{salt}$) in a 1 µm radius solid particle as a function of the molecular weight ($M_w$) of each aminium sulphate. Symbols: ● – MMAS; ■ – DMAS; ♦ – TMAS; ○ – MEAS; □ – DEAS; ◇ – TEAS.**





**Figure 5:** $n_{water}/n_{solute}$ **vs.** $a_w$ **plots for the six aminium sulphates. Symbols: black circles – CK-EDB comparative kinetics measurements; open circles – Sauerwein et al. (2015); grey dots – Clegg et al. (2013); line – (NH$_4$)$_2$SO$_4$ calculation from E-AIM model.**





**Figure 6: Osmotic coefficients ($\phi_{st}$) vs. square root of sulphate mass fraction ($m(SO_4^{2-})^{0.5}$) for the six aminium sulphates. Symbols: black circles – CK-EDB comparative kinetics measurements; open circles – Sauerwein et al. (2015); grey circles – Clegg et al. (2013); solid line – $(NH_4)_2SO_4$ calculation from E-AIM model; dashed lines – uncertainty of the osmotic coefficients for $(NH_4)_2SO_4$ corresponding to an error in $a_w$ of ±0.002, included to provide a guide as to the level of expected error in the osmotic coefficient with varying molality.**





**Figure 7: Mass of H₂O (kg) per mole of solute as a function of the degree of neutralisation of sulphuric acid by amine ($x_{salt}$). Symbols: solid circles – data from this work; open circles – data from Sauerwein et al. (2015). Lines: black – ZSR fitting of the data in Sauerwein et al. (2015); red dashed – same ZSR fitting but including the CK-EDB data point for each $a_w$; shaded envelopes – uncertainty associated to the fitting.**





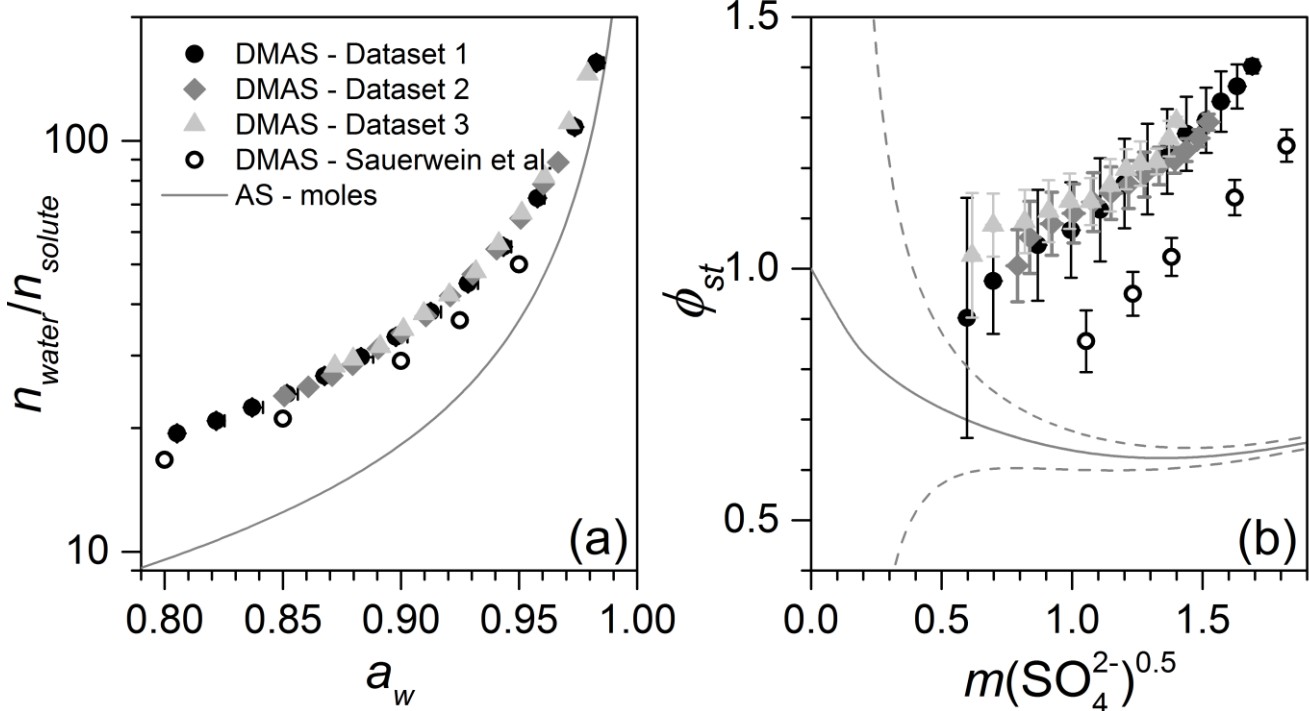

**Figure 8: Evaluation of the reproducibility of measurements of the hygroscopic properties of DMAS. Symbols: black dots, dark grey diamonds, light grey triangles – different datasets measured with the CK-EDB from DMAS solutions obtained indipendently; open circles – data from water activity measurements in Sauerwein et al. (2015); solid lines – (NH₄)₂SO₄ calculations from the E-AIM model; dashed lines – uncertainty on (NH₄)₂SO₄ osmotic coefficients corresponding to an error in $a_w$ of ±0.002.**



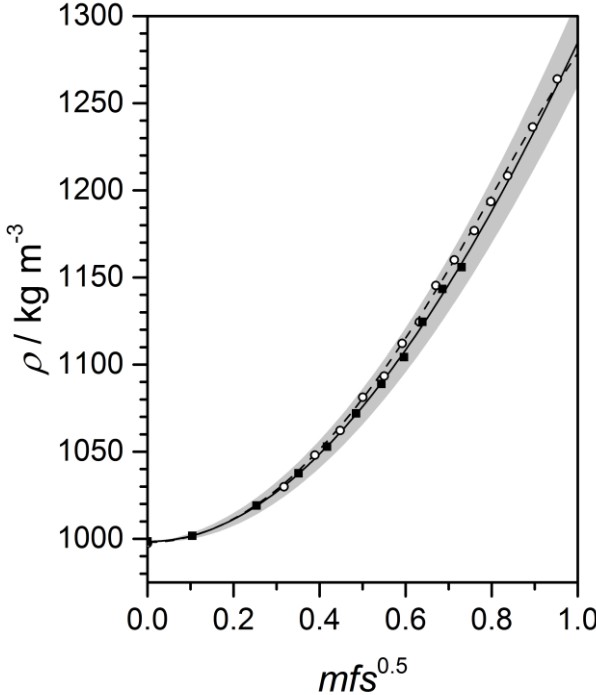

**Figure 9: DMAS density (ρ) dependence on mass fraction ($mfs^{0.5}$) of solute. Symbols: black squares – measured densities, this work; solid line – 3rd order polynomial fit of measured ρ values; grey shaded area – evaluated uncertainty of the density parameterisation (calculated as discussed in the main text); open circles – measured densities from Clegg et al. (2013); dashed line – densities from the apparent molar volumes fitting by Clegg et al. (2013).**





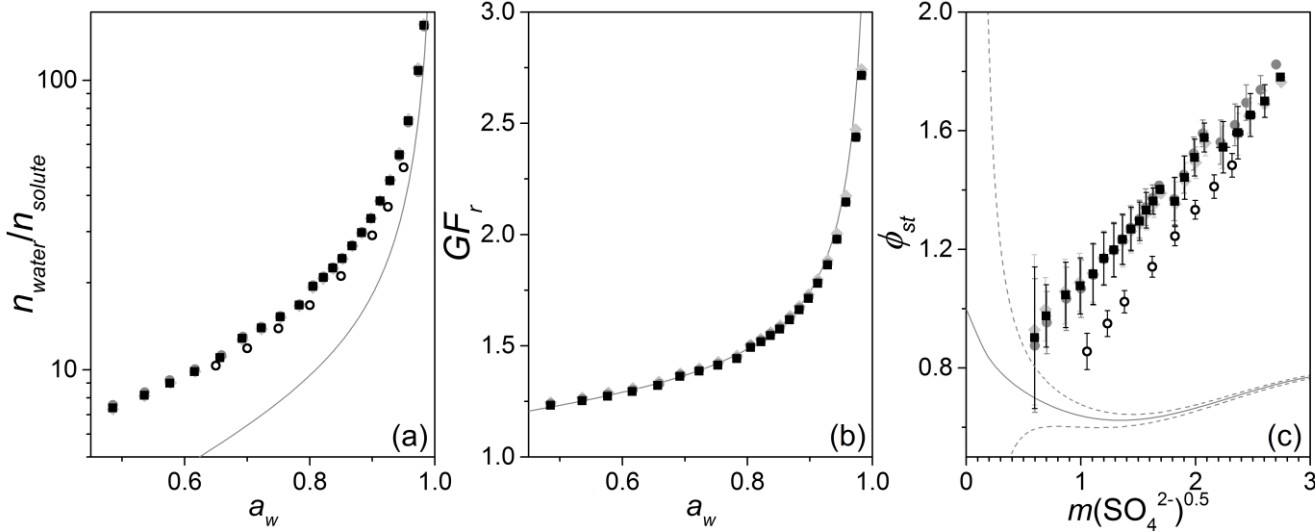

**Figure 10: Hygroscopic properties of DMAS obtained from the treatment of the CK-EDB kinetics evaporation data with the three different set of parameters in Table 1 for treating solution density. Symbols: black squares – original data; dark grey dots – obtained with 'Error+' parameters; light grey diamonds – obtained with 'Error-' parameters; open circles – Sauerwein et al. (2015); solid lines – E-AIM model for $(NH_4)_2SO_4$; dashed lines – uncertainty on $(NH_4)_2SO_4$ osmotic coefficients corresponding to an error in $a_w$ of ±0.002.**