# Peer review of "Hygroscopic Properties of Aminium Sulphate Aerosols"

_Atmospheric Chemistry and Physics, 2016_

## Referee Comment (RC1) · Anonymous Referee #1 · 13 Dec 2016

Rovelli et al. describes the hygroscopic properties of single aerosol particles composed of aminium sulfate using a cylindrical electrodynamic balance. They demonstration this is a good way to measure the aminium sulfate aerosol hygroscopicity due to fast measurement time that does not allow for evaporation of analyte. In this paper it is shown that there is a correlation between the nwater/nsolution dependent on the number of alkyl groups present on the amine. The results are compared to previous studies that have used other methods (bulk and tandem DMA) are discussed to describe where the differences from other methods could be derived. The experiments and attention to detail on the preparation of the solutions to avoid ambiguity are well thought through to eliminate sources of error. The paper is well written and discusses/compares results with previous literature in detail about discrepancies that are well explained through the discussion.

[Figure]

Specific Comments:

Pg. 2 Line 32: It would be nice to give numbers for seasonal variation here since other numbers are given. Additionally, on P3 line 3 mention the location of the measurements for completeness.

Pg. 3 line 5: It would be good to discuss the new particle formation in more detail with references.

Pg. 2 line13: Include the solubility of the amines compared to the salt products.

Pg. 4 line 23: Could you give the RHs used in this experiment here? The only place I saw them was in Figure 1 for the AS experiment.

Pg. 6 line 24: What is the difference in any calculations due to less of the amine (e.g. formation of aminium bisulphate) possibly being present?

Pg. 8 line 5: Mention error again here in regard to the TEAS concentration measurement since you are making a direct comparison and caution that there might be error.

Pg. 11 line 5: It would be nice to include the trend from Saurwein 2015 somewhere in reference to the GFr and nwater/nsolute since it is not discernable from the comparison graphs in Fig. 5. Are they the same trend as your measurements?

Pg. 13 line 15: Section 3.3 (and corresponding fig. 8) does not seem to add to the paper since the method has already been validated in a previous publication and could be moved to the supplemental information.

Figure 1: Note in 'experimental data graph' that you alternate between the sample and a standard (or clarify when it is performed). Some additional description of a, b, c, in the figure caption would be helpful in discerning the use of the figure even though it is described well in text.

Figure 3: Why is there no data for MMAS and TMAS between ~0.7-0.8 aw? What is the RH range used for these measurements?

Fig. 3,5,6, 8, 10: In description make it explicit that the ammonium sulfate is a model that you already validated earlier in the paper (Fig. 2). On first read through I missed this and was wondering why it was compared to a model and not the experiment.

Fig. S2: Why is the Saurwein et al. 2015 data not included here?

Technical Corrections:

Pg. 2 line 14: 'Because of this...' The 'this' is ambiguous and seemingly refers to solubility, reword sentence for clarity.

P6 line 15: Change the mass fraction to wt% for consistency within paper

Figures: Keep the lettering (a,b, etc.) capitalization consistent, Fig. 3 and S1 are capitalized while others are lowercase.

Fig. 5/6: note that the Clegg et al. 2013 data is from Qui and Zhang 2012, this was not clear.

Figures 5, 6, S2, S3: Include lettering in the graphs (a,b etc.) for consistency and ease of reference in text.

[Figure]

---

## Referee Comment (RC2) · Anonymous Referee #2 · 19 Dec 2016

This paper describes a new method of CK-EDB which can determine the hygroscopic properties of aerosol particles. In order to validate the method, the authors provided reproducible data for hygroscopic growth factor over the wide range of water activity. The results shown here agreed well with the results of previous studies that were performed with different methods. The manuscript presents in a clear, concise, and well-structured way, but I am afraid whether this paper is suitable for the scope of Atmos. Chem. Phys publication or not. Since the main focus of this work is to validate the new method, it would be better to be in a technical journal. If the authors would still like to publish the manuscript to ACP, the authors should address and implement my comments as below.

Major comments: The measurements of hygroscopic properties of six aminium sulfate aerosols over the water activity range of 0.5 ~ 1.0 are in remarkably good agreement

with the calculations and the previous studies. However, I do wonder to where/for what we could apply these results? To understand why this work is important in the area of atmospheric chemistry and physics, please describe atmospheric implications in more detail with a separate section.

Minor comments:

1. Page 4, line 23, Please add the RH and temperature values.

2. Page 6, line 13, Remove the comma after "mixed, "

3. Page 6, line 18, Please state at what temperature the amine solution was kept in an ice bath.

4. Page 8, line 12, Cite only once (Qiu and Zhang, 2012)

5. I wonder why there is no data point for MMAS and TMAS from ∼0.7 to ∼0.8 aw in Fig. 3.

---

## Referee Comment (RC3) · Anonymous Referee #3 · 19 Dec 2016

In this paper Grazia et al. describe the use of a Comparative Kinetic Electrodynamic Balance for investigation of the hygroscopic properties of Aminium Sulfate aerosols in comparison with water and Sodium Chloride solution drops with the same system and data collected by a number of other studies and methods. The experiments presented within are well thought out, the uncertainties in the data have been well investigated, and the results thoroughly compared with pervious data. However, I feel the tie to atmospheric chemistry, although present, is lacking. To be accepted for publication the authors should present more of a link to atmospheric processes and clearer indications of where this data this data will be most useful.

Below are more detailed comments on the manuscript.

Page 2 line 30: Rework the sentence starting with 'Their ambient conditions . . .'

[Figure]

Introduction overall: This is meant to bring everyone up to speed but I found it lacking. Consider including more information on atmospheric relevance, in situ particle formation, etc. For instance your mention of aminium sulfates role in cloud particle nucleation seems forced. If you go to the Lavi et al. paper you find right in their abstract that "Alkyl aminium sulfates have been postulated to constitute important components of nucleation and accumulation mode atmospheric aerosols." and "We infer that these species have very high CCN activity . . ."

Page 4 Line 6: It would be of benefit to the reader if a diagram of the EDB were included in the paper.

Page 4 line 16: How are you controlling the RH?

Page 4 line 22: You never explicitly state the range of RH and temperatures you are conducting your experiments at, over. Please include for completeness and link to atmospherically relevant conditions.

Page 4 line 26: You mention it once (on the noted line) but I think it would help to clarify that your droplets are alternatingly injected into the system as it is possible to have multiple particles or drops trapped in an EDB simultaneously. One question I had – You're residence timescale for a single particles is less than 30 s so the RH it is exposed to is arguably constant, but to what degree does the RH change over the course of the 10+ particle runs?

Page 8 line 13: I've seen this throughout the paper: '. . . estimated by Qiu and Zhang (2012) (Qiu and Zhang, 2012) is . . .' You have essentially cited the paper twice and the second citation should be removed. Other instances can be found on page 3 line 3 and page 15 line 29. There may be others I missed.

Page 12 line 19: I think this is the first time you introduce ZSR, make this acronym explicit in line 16 where you introduce Zdanovskii-Stokes-Robinson expression.

Page 15 line 22: Quantify fine variations.

[Figure]

Page 16 last paragraph: This doesn't seem to fit. Your previous paragraph starts with 'As a final remark . . .' then this is thrown in. This goes back to my main issue of making the paper more atmospherically relevant.

Figure 4: You can remove legend from caption text.

Figures 5 & 6: Consider labeling your subplots a-f as done in other multi-plot figures.

---

## Author Comment (AC1) · 10 Feb 2017

Grazia Rovelli[1,2], Rachael E.H. Miles[1], Jonathan P. Reid[1], Simon L. Clegg[3]

[1] School of Chemistry, University of Bristol, Bristol, BS8 1TS, UK
[2] Department of Earth and Environmental Sciences, University of Milano-Bicocca, 20124 Milan, Italy
[3] School of Environmental Sciences, University of East Anglia, Norwich NR4 7TJ, UK

*Correspondence to*: Jonathan P. Reid (j.p.reid@bristol.ac.uk)

**Response to Anonymous Referee #1**

The authors would like to thank Anonymous Referee #1 for their generally positive comments on the manuscript. We respond to the specific comments made by the referee below and identify the changes we have to the manuscript.

*Specific Comments:*

***Anonymous Referee #1:*** *Pg. 2 Line 32: It would be nice to give numbers for seasonal variation here since other numbers are given. Additionally, on P3 line 3 mention the location of the measurements for completeness.*

**Response:** Line 32 (Page 2) has now been expanded and the revised version reads as follows:

"The ambient concentrations of amines in the gas phase can span wide ranges, depending on the sampling location. For example, concentrations can be up to 140 mg m$^{-3}$ close to a city market (Namieśnik et al., 2003), 110-300 ng m$^{-3}$ in the exhaust gas of a waste disposal site (Kallinger and Niessner, 1999), and of the order of tens of μg m$^{-3}$ inside livestock buildings (Kallinger and Niessner, 1999). Concentrations also depend on the season: for example, single amines in the gas phase at a rural site in Turkey have been reported to be in the range 0.92-7.4 ng m$^3$ in Winter and 0.29-5.16 ng m$^3$ in Summer (Akyüz, 2008). Further, Pratt et al. (2009) measured a seasonal volatility dependence of alkylamines depending on the particles pH that affects the amines concentrations in the gas phase. In the condensed phase, amines can account for hundreds of pg m$^{-3}$ or a few ng m$^{-3}$ of aerosol mass."

The reviews by Ge at al. referenced on Line 3 (Page 3) include data for a large number of studies. Mentioning all the location of the measurements included in Ge's reviews is probably out of the scope of the overview given here. However, for clarity "at a large number of rural and urban environments" has been added (Page 3), now reading:

"An extensive review of the measured concentrations of a large number of amines both in the gas phase and in aerosols at a great variety of rural and urban environments is given by Ge et al. (2011a, 2011b)."

***Anonymous Referee #1:*** *Pg. 3 line 5: It would be good to discuss the new particle formation in more detail with references.*

**Response:** We agree with Referee #1 that the role of amines in new particles formation should be discussed in more detail and therefore we added a paragraph at line 5 (Page 3).

"The role of amines in new particles formation and growth has been highlighted by computational studies (DePalma et al., 2012; Loukonen et al., 2010; Ortega et al., 2012), as well as by laboratory (Almeida et al., 2013; Wang et al., 2010b) and field measurements (Kulmala et al., 2013; Mäkelä et al., 2001; Smith et al., 2010). As an example, trimethylamine was found to enhance the formation and growth of new particles (Wang et al., 2010a) because of the favourable heterogeneous neutralisation reactions between the amine gaseous molecules and $H_2SO_4$-$H_2O$ clusters. Smith et al. (2010) found variable but considerable concentrations of protonated amines in nanoparticles (8-10 nm diameter) during new particles formation events (47% of detected positive ions at an urban site in Mexico, 23% at remote site in Finland and 10-35% at sampling sites in Atlanta and Boulder). Since newly formed secondary particles are estimated to contribute to 45% of cloud condensation nuclei (CCN) (Merikanto et al., 2009), the presence of amines in CCN and their hygroscopic properties need to be taken into account to improve our understanding of the indirect effects of aerosol particles on climate (McFiggans et al., 2005), a key motivator for providing refined characterisation of the hygroscopic growth of aminium salt particles in this work."

***Anonymous Referee #1:*** *Pg. 2 line13: Include the solubility of the amines compared to the salt products.*

**Response:** The solubility of both the unprotonated amines (data reworked from Ge et al. 2011b) and of five aminium sulphates (data from Clegg et al. 2013) has now been included at page 3.

"The formation of aminium sulphates and other similar aminium salts increases the solubility of short-chained alkylamines from 7-45 wt% for the unprotonated form (calculation from data in Ge et al. (2011b)) to 84-91 wt% for five aminium sulphates in Clegg et al. (2013). This, correspondingly, increases their partitioning from the gas to the condensed phase (Barsanti et al., 2009; Yli-Juuti et al., 2013)."

***Anonymous Referee #1:*** *Pg. 4 line 23: Could you give the RHs used in this experiment here? The only place I saw them was in Figure 1 for the AS experiment.*

**Response:** This information has now been added at line 16, page 4.

"Temperature and gas phase RH ranges that are accessible with this experimental setup are -25 to 50 °C and 0 to 99%, respectively. All the comparative evaporation kinetics measurements presented here were performed at 20 °C and at gas phase RH values between ~50-90%."

***Anonymous Referee #1:*** *Pg. 6 line 24: What is the difference in any calculations due to less of the amine (e.g. formation of aminium bisulphate) possibly being present?*

**Response:** Although we have considered that the partial formation of aminium bisulphate could occur, this would increase the discrepancy that already exists between our data and that of Sauerwein et al. (2015) (for example, see Figure 7). Considerable care is taken to ensure that stoichiometric amounts of the sulphuric acid are added to the amine solution.

***Anonymous Referee #1:*** *Pg. 8 line 5: Mention error again here in regard to the TEAS concentration measurement since you are making a direct comparison and caution that there might be error.*

**Response:** We agree with Anonymous Referee #1 that mentioning again the possible error on TEAS is appropriate. The following sentences have therefore been added on page 8.

"As indicated in Section 2.2, it is worth reiterating that the results presented here and below for TEAS need to be interpreted cautiously because of the uncertainty of the TEA stock solution. However, the trends that have been observed for TEAS when compared to the other five aminium sulphate systems seem to be completely plausible and this may indicate that the assumed initial TEA concentration is reasonable."

***Anonymous Referee #1:*** *Pg. 11 line 5: It would be nice to include the trend from Saurwein 2015 somewhere in reference to the GFr and nwater/nsolute since it is not discernable from the comparison graphs in Fig. 5. Are they the same trend as your measurements?*

**Response:** We agree with Anonymous Referee #1 that from Figures 5 and 6 it is not possible to discern whether or not the trends observed in our work and those from Sauerwein et al. (2015) are similar. However, we already show the comparative trends from our measurements for both the methyl and ethylaminium sulphates series in Figure 3. Very similar trends are seen in the Sauerwein et al. data and we explicitly state on page 11: "However, despite the discrepancies that do exist, results from the two different experimental methods show the same qualitative hygroscopicity trends over the methyl and ethylaminium sulphates series". As a consequence, we do not feel that this merits providing an additional figure.

***Anonymous Referee #1:*** *Pg. 13 line 15: Section 3.3 (and corresponding fig. 8) does not seem to add to the paper since the method has already been validated in a previous publication and could be moved to the supplemental information.*

**Response:** We thank the referee for this comment: it has suggested to us that the actual aim of this paragraph was not clearly stated and this has led to some confusion. The main aim of the assessment of reproducibility was not to provide an evaluation of the CK-EDB approach; as Anonymous Referee #1 states, this has already been reported. Instead, the aim of Section 3.3 is to report on the reproducibility of the solution preparation procedures (described in Section 2.2) to ensure that the discrepancies with the results of Sauerwein et al. (2015) are not derived from our the sample preparation. We have added a few sentences in Section 3.3 and improved the clarity of the paragraph to ensure the purpose of the section is more obvious to the reader.

"A full validation of the retrieval of the hygroscopic properties of single trapped solution droplets from CK-EDB experiments has already been presented in a previous publication (Rovelli et al. 2016), where we demonstrated the accuracy of the approach by reporting hygroscopicity measurements for well-characterised inorganic components. In this Section, we evaluate the reproducibility of the solution preparation method (Sect. 2.2) with the aim of demonstrating that the results presented in the previous Sections are not affected by any random error associated with our approach for making the sample solutions. Data obtained from three different datasets of evaporating DMAS solution droplets are compared in Fig. 8. The variation in $n_{water}/n_{solute}$ with $a_w$ (Panel (a)) and the change in osmotic coefficient with square root of sulphate mass fraction (Panel (b)) are calculated from three different data sets, each arising from measurements with 10 droplets and prepared from different starting stock solution of diethylaminium sulphate, obtained separately one from the other by mixing DEA and $H_2SO_4$

in different days. Data from Sauerwein et al. (2015) (open circles) and calculations for ammonium sulphate (E-AIM model, line) are shown for comparison. Measurements were made with droplet evaporation into a gas phase RH of ~80%. The reproducibility of the data is very satisfactory, giving further evidence that the applied solution preparation procedure coupled to the retrieval of the hygroscopic properties with CK-EDB experiments is reliable. In addition, this consistency in experimental reproducibility strongly suggests that the discrepancies with the data of Sauerwein et al. (2015) do not originate from random errors associated with the CK-EDB experiments."

*Anonymous Referee #1: Figure 1: Note in 'experimental data graph' that you alternate between the sample and a standard (or clarify when it is performed). Some additional description of a, b, c, in the figure caption would be helpful in discerning the use of the figure even though it is described well in text.*

**Response:** Some more detail is now given in the figure caption and a legend has been added in panel (a) in order to make clearer that sample and probe droplets are alternated.

"Figure 1: Schematics representing the retrieval of hygroscopic growth curves from comparative kinetics experiments in a CK-EDB. An experimental sequence of alternating single evaporating probe and sample droplets is collected (a). For each pair of probe and sample droplets, the gas phase RH is inferred from the evaporation kinetics of the probe (b) and this information is used to analyse the corresponding sample droplet hygroscopic properties as indicated in (c)."

*Anonymous Referee #1:* Figure 3: Why is there no data for MMAS and TMAS between _0.7-0.8 aw? What is the RH range used for these measurements?

**Response:** Measurements were taken for a large number of systems over a wide range of conditions and it became apparent only later that measurements were not available for this small range of conditions for two compounds. We concluded that the trends were sufficiently clear that additional measurements were not essential.

*Anonymous Referee #1: Fig. 3,5,6, 8, 10: In description make it explicit that the ammonium sulfate is a model that you already validated earlier in the paper (Fig. 2). On first read through I missed this and was wondering why it was compared to a model and not the experiment.*

**Response:** To make this clear, we have added a line in the caption of Figure 3 to explicitly say this.

"Note that only E-AIM predictions for $(NH_4)_2SO_4$ are reported in this and subsequent figures with the CK-EDB measurements and the E-AIM model compared in Figure 2."

*Anonymous Referee #1: Fig. S2: Why is the Saurwein et al. 2015 data not included here?*

**Response:** Data from Sauerwein et al. (2015) come from bulk water activity measurements and, hence, we do not include them in Figure S2 ($GF_r$ vs. $a_w$ plots). In principle, one could convert the data from Sauerwein et al. (2015) into radial growth factors, but we consider that this would not really add much to the discussion in this work.

***Anonymous Referee #1:*** *Pg. 2 line 14: 'Because of this. . .' The 'this' is ambiguous and seemingly refers to solubility, reword sentence for clarity.*

**Response:** This sentence has now been reworked as follows:

"However, the physicochemical properties of aminium sulphates are much less well characterised than their inorganic counterpart, $(NH_4)_2SO_4$, even though they can play a fundamental role in the nucleation and growth of new particles (DePalma et al., 2012; Loukonen et al., 2010; Wang et al., 2010) and in cloud formation (Lavi et al., 2013)."

***Anonymous Referee #1:*** *P6 line 15: Change the mass fraction to wt% for consistency within paper.*

**Response:** The mass fraction at line 15, page 6, has now been converted to weight percentage.

*Anonymous Referee #1: Figures: Keep the lettering (a,b, etc.) capitalization consistent, Fig. 3 and S1 are capitalized while others are lowercase.*

**Response:** The capitalisation of the letters in all the Figures is now uniform.

***Anonymous Referee #1:*** *Fig. 5/6: note that the Clegg et al. 2013 data is from Qui and Zhang 2012, this was not clear.*

**Response:** The data plotted in Figure 5 is actually taken from Clegg et al. (2013), who reworked the radial growth factor original H-TDMA measurements from Qiu and Zhang (2012). The same is true for Figure 6, where we converted the data from Figure 5 in terms of osmotic coefficients. For the sake of clarity, we now state in the caption that the data from Clegg et al. (2013) are based on H-TDMA measurements from Qiu and Zhang (2012).

***Anonymous Referee #1:*** *Figures 5, 6, S2, S3: Include lettering in the graphs (a,b etc.) for consistency and ease of reference in text.*

**Response:** Letters have now been included in the various panels in order to make references in the text clearer.

---

## Author Comment (AC2) · 10 Feb 2017

Grazia Rovelli[1,2], Rachael E.H. Miles[1], Jonathan P. Reid[1], Simon L. Clegg[3]

[1] School of Chemistry, University of Bristol, Bristol, BS8 1TS, UK
[2] Department of Earth and Environmental Sciences, University of Milano-Bicocca, 20124 Milan, Italy
[3] School of Environmental Sciences, University of East Anglia, Norwich NR4 7TJ, UK

*Correspondence to*: Jonathan P. Reid (j.p.reid@bristol.ac.uk)

**Response to Anonymous Referee #2**

The authors would like to thank Anonymous Referee #2 for their comments on the manuscript. We respond to the specific comments made by the referee below and identify the changes we have to the manuscript.

*Anonymous Referee #2: This paper describes a new method of CK-EDB which can determine the hygroscopic properties of aerosol particles. In order to validate the method, the authors provided reproducible data for hygroscopic growth factor over the wide range of water activity. The results shown here agreed well with the results of previous studies that were performed with different methods. The manuscript presents in a clear, concise, and well-structured way, but I am afraid whether this paper is suitable for the scope of Atmos. Chem. Phys publication or not. Since the main focus of this work is to validate the new method, it would be better to be in a technical journal. If the authors would still like to publish the manuscript to ACP, the authors should address and implement my comments as below.*

*Major comments: The measurements of hygroscopic properties of six aminium sulfate aerosols over the water activity range of 0.5 _ 1.0 are in remarkably good agreement with the calculations and the previous studies. However, I do wonder to where/for what we could apply these results? To understand why this work is important in the area of atmospheric chemistry and physics, please describe atmospheric implications in more detail with a separate section.*

**Response:** In order to address the concerns of Referee #2 about the atmospheric relevance of this study, we have added some considerable detail in the Introduction section about the different processes aminium sulphates are involved in, together with a number of references to the literature. In particular, we added more detail about the role of gaseous amines molecules in the formation of new particles. In addition, we now clearly stress the fact that nanoparticles deriving from such new particles formation events and containing aminium sulphates have the potential ability to act as cloud condensation nuclei (CCN). For this reason, investigating and quantifying precisely the hygroscopic properties of aminium sulphates is particularly atmospherically relevant, since this information is valuable in understanding of the role of such compounds in cloud activation and therefore in the indirect effects of atmospheric aerosols on climate. Please refer to the Introduction (specifically the modifications already requested by Referee #1) and to the Summary and Conclusions (section 4), which is now titled "Atmospheric Importance and Conclusions". To highlight the atmospheric relevance, we now write at the beginning of Section 4:

"Quantifying the hygroscopic properties of aminium sulphates is important for understanding and modelling of the atmospheric processes in which they are involved. In particular, the role of short-chained alkylamines in the formation of new particles has been investigated in recent literature studies and found to be significant (Section 1). Aminium sulphate-rich nanoparticles that derive from new particles formation events can potentially act as CCN, and their hygroscopic properties must be well-characterised with the aim of reducing the overall uncertainties that currently affect our understanding of the indirect effects of atmospheric aerosols on climate. Robust and accurate data are essential for improving microphysical models of aerosol hygroscopicity; this study presents an extensive data set for an homologous series of six compounds, compared to ammonium sulphate, extending over a wide range in RH. In addition, it represents the most comprehensive characterisation of the hygroscopic response of aminium sulphate aerosol so far, complementing previous bulk phase measurements (comparable in accuracy but limited to higher water activity) and aerosol measurements at lower RH (with lower accuracy than achieved here). Previously, the bulk and aerosol measurements reported in the literature were in disagreement. Here, we report aerosol measurements that are in good agreement with the previously most accurate bulk phase data, resolving this discrepancy."

In addition, we already state later in this section the significance of these new data when compared with the earlier bulk phase data, stating:

"The main differences in approaches are that: we perform aerosol measurements that cover a wider range in water activity as compared with the bulk measurements of Sauerwein et al. (2015); and we provide direct measurement at amine-to-sulphate ratios of exactly 2:1, whereas Sauerwein et al. (2015) performed a ZSR fitting on data from solutions with variable amine-to-sulphates ratios and extrapolated water content for the exact 2:1 ratio. These new CK-EDB measurements suggest a higher level of hygroscopic growth for the aminium sulphates than previously reported by Sauerwein and co-workers when inferred from measurements over a range of amine-to-sulphates ratios; we have provided a refined parameterisation for all compositions."

***Minor comments***

***Anonymous Referee #2:*** *1. Page 4, line 23, Please add the RH and temperature values.*

**Response:** This information has now been added at line 16, page 4.

"Temperature and gas phase RH ranges that are accessible with this experimental setup are -25 to 50 °C and 0 to 99%, respectively. All the comparative evaporation kinetics measurements presented here were performed at 20 °C and at gas phase RH values between ~50-90%."

***Anonymous Referee #2:*** *2. Page 6, line 13, Remove the comma after "mixed, "*

**Response:** This comma has now been removed.

***Anonymous Referee #2:*** *3. Page 6, line 18, Please state at what temperature the amine solution was kept in an ice bath.*

**Response:** We have now explicitly said that the temperature was 0 °C.

"During both the titration of the amine stock solution with HCl and the preparation of the aminium sulphates solutions with $H_2SO_4$, the amine solution was kept in an ice bath (0 °C) and the addition of the acid was performed slowly and dropwise, in order to dissipate the heat generated by the neutralisation reaction and to avoid any possible amine volatilization."

*Anonymous Referee #2:* 4. Page 8, line 12, Cite only once (Qiu and Zhang, 2012)

**Response:** The double citation has now been removed.

*Anonymous Referee #2:* 5. I wonder why there is no data point for MMAS and TMAS from _0.7 to _0.8 aw in Fig. 3.

**Response:** We have responded to this issue in our response to the comments made by Referee #1. Measurements were taken for a large number of systems over a wide range of conditions and it became apparent only later that measurements were not available for this small range of conditions for two compounds. We concluded that the trends were sufficiently clear that additional measurements were not essential.

---

## Author Comment (AC3) · 10 Feb 2017

Grazia Rovelli[1,2], Rachael E.H. Miles[1], Jonathan P. Reid[1], Simon L. Clegg[3]

[1] School of Chemistry, University of Bristol, Bristol, BS8 1TS, UK
[2] Department of Earth and Environmental Sciences, University of Milano-Bicocca, 20124 Milan, Italy
[3] School of Environmental Sciences, University of East Anglia, Norwich NR4 7TJ, UK

*Correspondence to*: Jonathan P. Reid (j.p.reid@bristol.ac.uk)

**Response to Anonymous Referee #3**

The authors would like to thank Anonymous Referee #3 for their helpful comments on the manuscript. We respond to the specific comments made by the referee below and identify the changes we have to the manuscript.

***Anonymous Referee #3:*** *In this paper Grazia et al. describe the use of a Comparative Kinetic Electrodynamic Balance for investigation of the hygroscopic properties of Aminium Sulfate aerosols in comparison with water and Sodium Chloride solution drops with the same system and data collected by a number of other studies and methods. The experiments presented within are well thought out, the uncertainties in the data have been well investigated, and the results thoroughly compared with pervious data. However, I feel the tie to atmospheric chemistry, although present, is lacking. To be accepted for publication the authors should present more of a link to atmospheric processes and clearer indications of where this data this data will be most useful. Below are more detailed comments on the manuscript.*

**Response:** A similar comment was made by Referee #2 to which we have already responded. More specifically, we have added some considerable detail in the Introduction section about the different processes aminium sulphates are involved in, together with a number of references to the literature. In particular, we added more detail about the role of gaseous amines molecules in the formation of new particles. In addition, we now clearly stress the fact that nanoparticles deriving from such new particles formation events and containing aminium sulphates have the potential ability to act as cloud condensation nuclei (CCN). For this reason, investigating and quantifying precisely the hygroscopic properties of aminium sulphates is particularly atmospherically relevant, since this information is valuable in understanding of the role of such compounds in cloud activation and therefore in the indirect effects of atmospheric aerosols on climate. Please refer to the Introduction (specifically the modifications already requested by Referee #1) and to the Summary and Conclusions (section 4), which is now titled "Atmospheric Importance and Conclusions". To highlight the atmospheric relevance, we now write at the beginning of Section 4:

"Quantifying the hygroscopic properties of aminium sulphates is important for understanding and modelling of the atmospheric processes in which they are involved. In particular, the role of short-chained alkylamines in the formation of new particles has been investigated in recent literature studies and found to be significant (Section 1). Aminium sulphate-rich nanoparticles that derive from new particles formation events can potentially act as CCN, and their hygroscopic properties must be

well-characterised with the aim of reducing the overall uncertainties that currently affect our understanding of the indirect effects of atmospheric aerosols on climate. Robust and accurate data are essential for improving microphysical models of aerosol hygroscopicity; this study presents an extensive data set for an homologous series of six compounds, compared to ammonium sulphate, extending over a wide range in RH. In addition, it represents the most comprehensive characterisation of the hygroscopic response of aminium sulphate aerosol so far, complementing previous bulk phase measurements (comparable in accuracy but limited to higher water activity) and aerosol measurements at lower RH (with lower accuracy than achieved here). Previously, the bulk and aerosol measurements reported in the literature were in disagreement. Here, we report aerosol measurements that are in good agreement with the previously most accurate bulk phase data, resolving this discrepancy."

In addition, we already state later in this section the significance of these new data when compared with the earlier bulk phase data, stating:

"The main differences in approaches are that: we perform aerosol measurements that cover a wider range in water activity as compared with the bulk measurements of Sauerwein et al. (2015); and we provide direct measurement at amine-to-sulphate ratios of exactly 2:1, whereas Sauerwein et al. (2015) performed a ZSR fitting on data from solutions with variable amine-to-sulphates ratios and extrapolated water content for the exact 2:1 ratio. These new CK-EDB measurements suggest a higher level of hygroscopic growth for the aminium sulphates than previously reported by Sauerwein and co-workers when inferred from measurements over a range of amine-to-sulphates ratios; we have provided a refined parameterisation for all compositions."

***Anonymous Referee #3:*** *Page 2 line 30: Rework the sentence starting with 'Their ambient conditions...'*

**Response:** This sentence has now been reworded as follow: "The ambient concentrations of amines in the gas phase can span wide ranges, depending on the sampling location. For example, concentrations can be up to 140 mg m$^{-3}$ close to a city market (Namieśnik et al., 2003), 110-300 ng m$^{-3}$ in the exhaust gas of a waste disposal (Kallinger and Niessner, 1999), and of the order of tens of µg m$^{-3}$ inside livestock buildings (Kallinger and Niessner, 1999).  Concentrations also depend on the season: for example,  single amines in the gas phase at a rural site in Turkey have been reported to be in the range 0.92-7.4 ng m$^3$ in Winter and 0.29-5.16 ng m$^3$ in Summer (Akyüz, 2008)."

***Anonymous Referee #3:*** *Introduction overall: This is meant to bring everyone up to speed but I found it lacking. Consider including more information on atmospheric relevance, in situ particle formation, etc. For instance your mention of aminium sulfates role in cloud particle nucleation seems forced. If you go to the Lavi et al. paper you find right in their abstract that "Alkyl aminium sulfates have been postulated to constitute important components of nucleation and accumulation mode atmospheric aerosols." and "We infer that these species have very high CCN activity . . ."*

**Response:** We thank the referee for this suggestion. As indicated in our response to their first comment, the Introduction section has now been reworked and expanded in order to make include more information on the atmospheric relevance of the investigation of the hygroscopic properties of aminium sulphates. In particular, we state on page 3:

"The role of amines in new particles formation and growth has been highlighted by computational studies (DePalma et al., 2012; Loukonen et al., 2010; Ortega et al., 2012), as well as by laboratory (Almeida et al., 2013; Wang et al., 2010b) and field measurements (Kulmala et al., 2013; Mäkelä et al.,

2001; Smith et al., 2010). As an example, trimethylamine was found to enhance the formation and growth of new particles (Wang et al., 2010a) because of the favourable heterogeneous neutralisation reactions between the amine gaseous molecules and $H_2SO_4$-$H_2O$ clusters. Smith et al. (2010) found variable but considerable concentrations of protonated amines in nanoparticles (8-10 nm diameter) during new particles formation events (47% of detected positive ions at an urban site in Mexico, 23% at remote site in Finland and 10-35% at sampling sites in Atlanta and Boulder). Since newly formed secondary particles are estimated to contribute to 45% of cloud condensation nuclei (CCN) (Merikanto et al., 2009), the presence of amines in CCN and their hygroscopic properties need to be taken into account to improve our understanding of the indirect effects of aerosol particles on climate (McFiggans et al., 2005), a key motivator for providing refined characterisation of the hygroscopic growth of aminium salt particles in this work."

*Anonymous Referee #3:* *Page 4 Line 6: It would be of benefit to the reader if a diagram of the EDB were included in the paper.*

**Response:** Because of the considerable length of the manuscript, and since 10 figures have already been included, we prefer not to include a diagram of the EDB instrument here; such diagrams already appear in our earlier papers. However, a schematic of the EDB setup has now been included in the supporting information and referred to in the text.

*Anonymous Referee #3:* *Page 4 line 16: How are you controlling the RH?*

**Response:** Different RHs are obtained by mixing a wet and a dry nitrogen flow in different ratios. This information has now been included on page 4. For completeness, how the temperature is controlled in this experimental setup has also been included. These details were not included in the original version of the manuscript, instead choosing to refer the reader to previous publications (Rovelli et al., 2016; Davies et al., 2013) for a detailed description.

"The gas flow RH is modified by mixing different ratios of a humidified and a dry nitrogen flow and is inferred from the evaporation kinetics of probe droplets, as described below in this section. The temperature within the trapping chamber is controlled by a circulating a 1:1 volume mixture of water and ethylene glycol, through the lid and the bottom of the chamber."

*Anonymous Referee #3:* *Page 4 line 22: You never explicitly state the range of RH and temperatures you are conducting your experiments at, over. Please include for completeness and link to atmospherically relevant conditions.*

**Response:** This information has now been added on page 4. Although a wide range in atmospherically relevant RHs is addressed in the measurements we present here, we have not yet performed a temperature dependence for these measurements. This will be addressed in a subsequent publication.

"Temperature and gas phase RH ranges that are accessible with this experimental setup are -25 to 50 °C and 0 to 99%, respectively. All the comparative evaporation kinetics measurements presented here were performed at 20 °C and at gas phase RH values between ~50-90%."

***Anonymous Referee #3:*** *Page 4 line 26: You mention it once (on the noted line) but I think it would help to clarify that your droplets are alternatingly injected into the system as it is possible to have multiple particles or drops trapped in an EDB simultaneously. One question I had – You're residence timescale for a single particles is less than 30 s so the RH it is exposed to is arguably constant, but to what degree does the RH change over the course of the 10+ particle runs?*

**Response:** To make sure that it is clear that all evaporation kinetics experiments are performed by alternating single probe droplets to single sample droplets, "single" (droplets) has been added at lines 15, 23, 24 (page 4) and "singly-trapped" (probe and sample droplets) has been added at line 28 (page4).

With respects to Referee #3's concerns about the stability of the gas phase RH over a run of 10+ particles, first we would like to point out that the evaporation kinetics of each sample droplet is analysed using the RH value coming from the fitting of the previous probe droplet. This ensures that even when there are very slight fluctuations in the gas phase RH, they would be taken into account in the retrieval of the hygroscopic properties of the sample droplets. That said, RH fluctuations over a typical run of 20 droplets (10 probe/10 samples) are of the order of 0.2% and never exceed 0.5% RH. In order to clarify this aspect, a few sentences have been added on page 5.

"RH fluctuations over the run of ten pairs or more of probe and sample droplets are very slight, typically of the order of 0.2% RH and never exceeding 0.5% RH. However, it should be noted that slight RH fluctuations are taken into account in our approach: the gas phase RH is monitored before every sample droplet by injecting a probe droplet and data from this probe droplet are directly used in the sample droplet evaporation analysis."

***Anonymous Referee #3:*** *Page 8 line 13: I've seen this throughout the paper: '. . . estimated by Qiu and Zhang (2012) (Qiu and Zhang, 2012) is . . .' You have essentially cited the paper twice and the second citation should be removed. Other instances can be found on page 3 line 3 and page 15 line 29. There may be others I missed.*

**Response:** Thank you for identifying this problem. The double references mentioned have now been removed and we have also checked throughout the manuscript to make sure there are no other double references remaining.

***Anonymous Referee #3:*** *Page 12 line 19: I think this is the first time you introduce ZSR, make this acronym explicit in line 16 where you introduce Zdanovskii-Stokes-Robinson expression.*

**Response:** We have added the definition of the acronym on page 12 where it is used for the first time.

***Anonymous Referee #3:*** *Page 15 line 22: Quantify fine variations.*

**Response:** 'Fine variations' at line 22, page 15, is now quantified as follows.

"Measurements from our new approach provide a level of accuracy that reveals clearly the fine variations in hygroscopic growth (down to discernible difference is $GF_r$ of order 0.01-0.02) that occur with molecular structure and substitution, and avoids the additional complexity of volatilisation of

semi-volatile components during hygroscopic growth with measurements complete in a matter of a few seconds."

**Anonymous Referee #3:** *Page 16 last paragraph: This doesn't seem to fit. Your previous paragraph starts with 'As a final remark . . .' then this is thrown in. This goes back to my main issue of making the paper more atmospherically relevant.*

**Response:** Thank you for this comment. We agree that having "As a final remark.." at the start of the second to last paragraph is a little misleading, so we have now changed this. However, we think that the considerations in the last paragraph are quite important, because this is the first one of a series of papers where we will report measurements of the hygroscopic properties of increasingly more complex organic and mixed inorganic-inorganic aerosol systems. We have addressed the question of atmospheric relevance earlier in our response.

**Anonymous Referee #3:** *Figure 4: You can remove legend from caption text.*

**Response:** The legend has now been removed from the caption.

**Anonymous Referee #3:** *Figures 5 & 6: Consider labeling your subplots a-f as done in other multi-plot figures.*

**Response:** Figures 5 and 6 are now labelled with a-f to indicate each subplot. References to these labels have also been included in the main text.